# Learn2Weight: Weights Transfer Defense against Similar-domain Adversarial Attacks

## Abstract

Recent work in black-box adversarial attacks for NLP systems has attracted attention. Prior black-box attacks assume that attackers can observe output labels from target models based on selected inputs. In this work, inspired by adversarial transferability, we propose a new type of black-box NLP adversarial attack that an attacker can choose a similar domain and transfer the adversarial examples to the target domain and cause poor performance in target model. Based on domain adaptation theory, we then propose a defensive strategy, called Learn2Weight, which trains to predict the weight adjustments for target model in order to defense the attack of similar-domain adversarial examples. Using Amazon multi-domain sentiment classification dataset, we empirically show that Learn2Weight model is effective against the attack compared to standard black-box defense methods such as adversarial training and defense distillation. This work contributes to the growing literature on machine learning safety.

## 1 Introduction

As machine learning models are applied to more and more real-world tasks, addressing machine learning safety is becoming an increasingly pressing issue. Deep learning algorithms have been shown to be vulnerable to adversarial examples (Szegedy et al., 2013; Goodfellow et al., 2014; Papernot et al., 2016a). In particular, prior black-box adversarial attacks assume that the adversary is not aware of the target model architecture, parameters or training data, but is capable of querying the target model with supplied inputs and obtaining the output predictions. The phenomenon that adversarial examples generated from one model may also be adversarial to another model is known as adversarial transferability (Szegedy et al., 2013).

Motivated by adversarial transferability, we conjecture another black-box attack pipeline where the adversary does not even need to have access to the target model nor query labels from crafted inputs. Instead, as long as the adversary knows the task of the target, he can choose a similar domain, to build a substitute model and then attack the target model with adversarial examples that are generated from the attack domain.

Figure 1: Diagrammatic representation of the problem

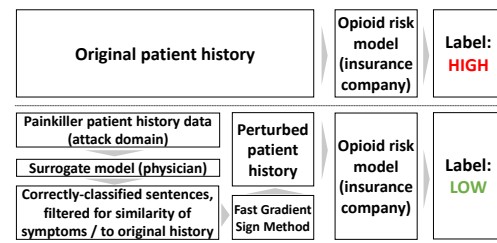

(a) Generalized architecture of similarity-based attacks.

(b) Flow of how an adversary physician can leverage similarity attack to fool opioid risk models.

The similar-domain adversarial attack may be more practical than prior blackbox attacks as label querying from target model is not needed. This attack can be illustrated with the following example (Figure 1b) in medical insurance fraud (Finlayson et al., 2019). Insurance companies may use hypothetical opioid risk models to classify the likelihood (high/low) of a patient to abuse the opioids to be prescribed, based on the patient's medical history as text input. Physicians can run the original patient history through the attack pipeline to generate an adversarial patient history, where the original is more likely to be rejected ("High" risk) and the adversarial is more likely to be accepted ("Low" risk). Perturbations in patient history could be, for example, a slight perturbation from "alcohol abuse" to "alcohol dependence", and it may successfully fool the insurance company's model.

Based on domain adaption theory (Ben-David et al., 2010), we conjecture that it is the domain-variant features that cause the success of the similar-domain attack. The adversarial examples with domain-variant features are likely to reside in the low-density regions (far away from decision boundary) of the empirical distribution of the target training data which could fool the target model (Zhang et al., 2019b). Literature indicates that worsened generalizability is a tradeoff faced by existing defenses such as adversarial training (Raghunathan et al., 2019) and domain generalization techniques (Wang et al., 2019). In trying to increase robustness against adversarial inputs, a model faces a tradeoff of weakened accuracy towards clean inputs. Given that an adversarial training loss function is composed of a loss against clean inputs and loss against adversarial inputs, improper optimization where the latter is highly-optimized and the former weakly-optimized does not improve general performance in the real-world. To curb this issue, methods have been proposed (Zhang et al., 2019b; Lamb et al., 2019; Schmidt et al., 2018), such as factoring in under-represented data points in training set.

To defend against this similar-domain adversarial attack, we propose a weight transfer network approach, **Learn2Weight**, so that the target model's decision boundary can adapt to the examples from low-density regions. Experiments confirm the effectiveness of our approach against the similar-domain attack over other baseline defense methods. Moreover, our approach is able to improve robustness accuracy without losing the target model's standard generalization accuracy.

Our contribution can be summarized as follows:

- We are among the first to propose the similar-domain adversarial attack. This attack pipeline relaxes the previous black-box attack assumption that the adversary has access to the target model and can query the model with crafted examples.

- We propose a defensive strategy for this attack based on domain adaptation theory. Experiment results show the effectiveness of our approach over existing defense methods, against the similar-domain attack.

## 2 RELATED WORK

Recent work in adversarial attack for NLP systems has attracted attention. See (Zhang et al., 2020) survey for an overview of the adversarial attack in NLP. Existing research proposes different attack methods for generating adversarial text examples (Moosavi-Dezfooli et al., 2016; Ebrahimi et al., 2018; Wallace et al., 2019). The crafted adversarial text examples have been shown to fool the state-of-the-art NLP systems such as BERT (Jin et al., 2019). A large body of adversarial attack research focuses on black-box attack where the adversary builds a substitute model by querying the target model with supplied inputs and obtaining the output predictions. The key idea behind such black-box attack is that adversarial examples generated from one model may also be mis-classified by another model, which is known as adversarial transferability (Szegedy et al., 2013; Cheng et al., 2019). While prior work examines the transferability between different models trained over the same dataset, or the transferability between the same or different model trained over disjoint subsets of a dataset, our work examines the adversarial transferability between different domains, which we call similar-domain attack.

| | Attack domain: baby, Target domain: books | |
|---|---|---|
| Original sentence (Actual label: Pos) | I purchased this toy for my son when he was 4 months old. At first, he seemed a little intimidated by the toys. | Pos (0.712) |
| Adversarial sentence | *I obtained this toys for my children when he was 4 weeks senior. At first, he hoped a modest harassed by the toy.* | Neg (0.364) |
| Original sentence (Actual label: Pos) | It felt like a big commitment for me to have to run the program 2 times a day, and near the end of my pregnancy I was annoyed with having anything strapped across my belly. | Pos (0.825) |
| Adversarial sentence | *It felt like a big committed for me to have to run the program 2 length a day, and near the end of my pregnancy I was annoyed with takes anything strapped across my belly.* | Neg (0.420) |
| | Attack domain: dvd, Target domain: baby | |
| Original sentence (Actual label: Pos) | Fast times at ridgemont high is a clever, insightful, and wicked film! It is not just another teen movie. | Pos (0.614) |
| Adversarial sentence | *Sooner days at ridgemont high is a sane, thoughtful, and wicked flick! It is not just another adolescent flick.* | Neg (0.335) |
| Original sentence (Actual label: Pos) | This dvd gives a very good 60 minute workout. As others have pointed out the cardio is very dancy. The first time I did it, I felt a bit awkward with the steps. | Pos (0.647) |
| Adversarial sentence | *This dvd gives a awfully okay 60 minute exercise. As others have pointed out the cardio is very dancy. The first time I did it, I perceived a bit awkward with the steps.* | Neg (0.258) |

Table 1: Comparison of attack domain sentences correctly classified when unperturbed by respective attack domain models and target domain models, then misclassified after perturbation by target models trained on **books** and **baby** domain. The perturbations are in blue, and prediction confidence in brackets.

## 3 SIMILAR-DOMAIN ADVERSARIAL ATTACK

### 3.1 ADVERSARIAL ATTACK BACKGROUND

Adversarial attacks modify inputs to cause errors in machine learning inference (Szegedy et al., 2013). We utilize the basic gradient-based attack method *Fast Gradient Sign Method (FGSM)* (Goodfellow et al., 2014) and its variants, *RAND-FGSM* (Tramèr et al., 2017) and *Basic Iterative Method (BIM)* (Kurakin et al., 2016a;b; Xie et al., 2018). Other NLP adversarial generation algorithms could also be used, such as DeepFool (Moosavi-Dezfooli et al., 2016), HotFlip (Ebrahimi et al., 2018), universal adversarial trigger (Wallace et al., 2019), and TextFooler (Jin et al., 2019). To perform gradient-based perturbations upon discrete space data, we follow (Yang et al., 2018) to generate adversarial text.

Our proposed similar-domain adversarial attack is in-variant to adversarial algorithm, meaning that the adversarial algorithm used would not affect the attack performance. Without losing generality, we denote $Adv(f, x)$ as an NLP adversarial text generation method, defined as below.

**Definition 1. NLP Adversarial Generation.** Given a deep neural network model $f$ built on text data $X$, an NLP adversarial generation method produces one adversarial instances $x' \leftarrow Adv(f, x)$ for $x \in X$, $x' \approx x$. The goal of the adversarial attack is to deviate the label to incorrect one $f(x') \neq f(x)$.

### 3.2 SIMILAR-DOMAIN ADVERSARIAL ATTACK

We present the architecture of similar-domain adversarial attack in Figure 1a. The defender, the target of the attack, constructs a target model trained on domain text data $T$ (0). An attacker, only having a rough idea about the target's task but lacking direct access to the target data or target model parameters, collects attack data from a similar domain $S$ and trains an attack model (1). He runs the attack model on the test data (2) to obtain correctly-classified instances (3). He chooses an adversarial attack algorithm and generates a set of adversarial samples $A$ (4). He exposes $A$ to the target model, hoping $A$ mislead the target model to produce an output of his choice (5). This type of attack works best as an adversarial attack that compromises systems that base decision-making on one-instance.

**Definition 2. Similar-domain Adversarial Attack.** A target model $f$, built on target domain data $T$, is a deep neural network model with parameter weights $W_T$ that maps a text instance to a label: $y \leftarrow f(X, W_T)$. An adversary chooses a source attack domain $S$, builds a substitute model $f_S$, and generates a set of adversarial examples $A$ from $S$ using $Adv(f_S, S)$, so that during an attack $f(A, W_T) = f_S(A)$.

| Target Domain | book | | | magazine | | | baby | | |
|---|---|---|---|---|---|---|---|---|---|
| Original Accuracy | 0.880 | | | 0.960 | | | 0.890 | | |
| Intra-attack Accuracy | 0.525 | | | 0.570 | | | 0.632 | | |
| Attack Domain | magazine | baby | dvd | baby | dvd | book | dvd | book | magazine |
| Unperturbed Accuracy | 0.726 | 0.646 | 0.745 | 0.739 | 0.663 | 0.673 | 0.624 | 0.652 | 0.665 |
| **After-attack Accuracy** | 0.398 | **0.421** | 0.395 | **0.381** | 0.366 | 0.343 | 0.365 | 0.386 | **0.401** |
| Shared Vocab | 0.381 | 0.255 | 0.455 | 0.260 | 0.345 | 0.381 | 0.270 | 0.255 | 0.260 |
| Transfer Loss | 0.017 | **0.071** | 0.000 | **0.079** | 0.022 | 0.010 | 0.066 | 0.050 | **0.069** |

Table 2: Similar-domain attack performance. **Bold** indicates the least successful attack domain (i.e. highest after-attack accuracy) for each target domain, as well as the corresponding transfer loss value.

### 3.3 DOMAIN SIMILARITY

Here, domain similarity refers to the similarity between attacker's chosen domain and defender's domain. **SharedVocab** measures the overlap of unique words, in each of the datasets; a higher degree of overlapping vocabulary implies the two domains are more similar. We also use **Transfer Loss**, a standard metric for domain adaptation Blitzer et al. (2007); Glorot et al. (2011), to measure domain similarity; lower loss indicates higher similarity. The test error from a target model trained on target domain $T$ and evaluated on attack domain $S$ returns transfer error $e(S, T)$. The baseline error $e(T, T)$ term is the test error obtained from target model trained on target domain (train) data $T$ and tested on target domain (evaluation) data $T$. This computes the transfer loss, $tf(S, T) = e(S, T) - e(T, T)$.

## 4 IS THE ATTACK EFFECTIVE?

### 4.1 SETUP

**Dataset.** We simulate the similar-domain adversarial attack using Amazon's multi-domain sentiment classification dataset (Blitzer et al., 2007), a commonly-used dataset in cross-domain sentiment classification[1], with 1,000 positive and 1,000 negative reviews for each of the 25 product categories.

**Model.** In practice, there could be unlimited choice for the attack model and target model, such as different deep learning architecture, different training parameters. To simplify the discussion, we choose *Long Short-Term Memory (LSTM)* network as a suitable baseline sentiment classification model (Wang et al., 2018) for our target model and attack model. The architecture consists of 64 LSTM cells, 80% dropout, using a $sigmoid$ activation function.

**Metrics.** We first report the accuracy of the target models on the target domain test samples before the attack as the *original accuracy*. Then we measure the accuracy of the target models against adversarial samples crafted from the attack domain samples, denoted as the *after-attack accuracy*. *Intra-attack accuracy* denotes the after-attack accuracy where the attack domain is identical to the target domain. By comparing original and after-attack accuracy, we can evaluate the success of the attack. The greater the gap between the original and after-attack accuracy, the more successful the attack. *Unperturbed accuracy* measures the accuracy of the target model against the complete, unperturbed test set of the attack domain, to demonstrate that any drop in classification accuracy is not from domain shift alone but from adversarial transferability.

### 4.2 RESULTS

The similar-domain adversarial attack results are presented in Table 2. We see a significant gap between original accuracy and after-attack accuracy indicating that this attack can impose valid threat to a target NLP system. After the similar-domain adversarial attack, the accuracy drops dramatically by a large margin. Take the book target domain for example, when the attack domain is magazine, the after-attack accuracy drops to 0.398, and when the attack domain is baby, the accuracy is 0.421. Moreover, we observe a positive correlation between transfer loss and after-attack accuracy, and a negative correlation between shared vocab and after-attack accuracy.

---

[1]Data is available from `https://www.cs.jhu.edu/~mdredze/datasets/sentiment/`

## 5 DEFENDING AGAINST SIMILAR-DOMAIN ADVERSARIAL ATTACK

In order to defend against a similarity based adversarial attack, it is critical to block adversarial transferability. Adversarial training is the most intuitive yet effective defense strategy for adversarial attack (Goodfellow et al., 2014; Madry et al., 2017). However, this may not be effective for two reasons. First, there is no formal guidance for generating similar-domain adversarial examples because the defender has no idea what the attack data domain is. Second, simple feeding the target model with adversarial examples may even hurt the generalization of the target model (Raghunathan et al., 2019; Zhang et al., 2019a; Su et al., 2018), which is also confirmed in our experiments.

### 5.1 WEIGHT TRANSFER LEARNING

The use of *weight transfer networks* (Ha et al., 2016; Hu et al., 2018; Kuen et al., 2019) is concerned with adapting weights from one model into another, and generating/predicting the complete set of weights for a model given the input samples. In our context, distinctly different weights are produced for target models trained on inputs of different domains, and feature transferability (Yosinski et al., 2014) in the input space can be expected to translate to weight transferability in the model weights space. Rather than completely regenerating classification weights, our model robustification defense, *Learn2Weight L2W* predicts the perturbation to existing weights ($w' = w + \Delta w$) for each new instance.

### 5.2 LEARN2WEIGHT MODEL

We conjecture that an effective defense strategy is to perturb the target model weights depending on the feature distribution of the input instance. $L2W$ (Algorithm 1) recalculates the target model weights depending on the input. $L2W$ (Algorithm 2) trains on sentences from different domains and a weight differential for that domain (the weight adjustment required to tune the target model's weights to adapt to the input's domain). We obtain the weight differential $\Delta W$ by finding the difference between the weights of $f$ trained on sentence:label pairs from a specific domain $W_{S_j}$ and weights of $f$ trained on sentence:label pairs from the target domain $W_T$. Other training models may be possible; here we trained a sequence-to-sequence network (Sutskever et al., 2014) on sentence:$\Delta W$ pairs.

| **Algorithm 1:** Learn2Weight: Inference | **Algorithm 2:** Learn2Weight: Training |
|---|---|
| **inference** $(X_i^{adv}, L2W(\cdot), f(W_T, T))$ 

     **Input** : Passed arguments include the adversarial input $X^{adv}$, the Learn2Weight model $L2W(\cdot)$, and the target model $f(W_T, T)$ 
     **Output** : Target model with weighted updated by Learn2Weight $f(W^*, T)$ is the expected output 

     Pass $X_i^{adv}$ as in input into the trained $L2W(\cdot)$ function, and the weight differential required for the target model with weights $W_T$ is $\Delta W$. 
     $\Delta \hat{W} \leftarrow L2W(X_i^{adv})$; 

     The returned function is the target model with updated weights $W_T + \Delta W$. 
     **return** $f(W_T + \Delta \hat{W}, T)$; | **train** $(T, D)$ 

     **Input** : Target domain $T = \{T_i\}_{i=0}^N$; Set of $M$ domains $D = \{S_{i,j}\}_{i,j=0}^{N,M}$; with $N$ sentences, $i$ indexing specific sentence tensor and $j$ indexing specific domains 
     **Output** : Trained Learn2Weight model $L2W(\cdot)$ 

     Initialize empty $X$ and $Y$ to store sentences $X_i$ from each domain $j$ with corresponding weight differential. 
     $X \leftarrow \emptyset; Y \leftarrow \emptyset$; 

     Compute weights of $f$ trained on $T$. 
     $W_T \leftarrow f(T)$; 
     $X \leftarrow T; Y \leftarrow W_T - W_T$; 

     Train each domain $S_j$, compute respective weights, append the differential $\Delta W$ to $Y$ and each sentence in $S_{i,j}$ into $X$. 
     **foreach** *domain $S_j \in D$* **do** 
         $W_{S_j} \leftarrow f(S_j)$; 
         $\Delta W \leftarrow W_{S_j} - W_T$; 
         $X \leftarrow S_j; Y \leftarrow \Delta W$; 

     Train a sequence model on $\{X : Y\}$ pairs, and return the model as Learn2Weight $L2W$. 
     **return** $L2W(X, Y)$; |

---

**Algorithm 3:** tf-optimization

---

**tfOptimization** $(T, M, n_{max})$

    **Input**   :Target domain $T = \{T_i\}_{i=0}^{N}$ to be used in synthesizing $M$ similar domains; with $N$ sentences,
                $i$ indexing specific sentence tensor; $n_{max}$ is the max number of tf-optimization iterations
    **Output**:Set $D$ containing $M$ domains

    Initialize empty $D$ to store synthesized domains $S_j$ of index $j$.
    $D \leftarrow \emptyset; j \leftarrow 0;$

    **while** $j < M$ **do**
        Run each iteration until $n_{max}$.
        **for** $iter \leftarrow 0$ **to** $n_{max}$ **do**
            Apply adversarial perturbations to $T$.
            $T_{iter}^{adv} \leftarrow Adv(f, T);$

            Determine change to $Adv(\cdot)$ or $iter$ depending on computed transfer loss.
            **if** $check(tf(T_{iter}^{adv}, T)) \leftarrow True$ **then**
                If low enough, $T_{iter}^{adv}$ can be added as synthetic domain into $D$.
                $D \leftarrow T_{iter}^{adv};$
                break;
            **else**
                Adjust perturbation parameters.
                $Adv(\cdot) \leftarrow adjust(Adv(\cdot));$
        $j \leftarrow j + 1;$
    Return set of Domains to be used for training by $L2W$.
    **return** $D$;

---

## 5.3   Transfer Loss Optimization

To generate synthetic domains of varying domain similarity so that defenders defend their model using only target domain data $T$, the following equation introduces **transfer loss optimization** (Algorithm 3). The defender iteratively generates adversarial examples $X_N^{adv}$ while maximizing the transfer loss function $tf$; this produces a *substitute attack domain* corpora $S_j$. We iteratively adjust perturbation parameters $Adv(\cdot)$ and iteration count $N$ to minimize the transfer loss of our generated dataset.

$$\underset{N, Adv(\cdot)}{\arg\min} tf_N\left(X_N^{adv}, T\right) = e\left(Adv(f, T), T\right) - e\left(T, T\right)$$

## 5.4   Explanation: Blocking Transferability

To facilitate our explanation, we adapt from domain adaptation literature (Ben-David et al., 2010; Liu et al., 2019; Zhang et al., 2019c):

$$e(A, T) \leq e(T, T) + d_{\mathcal{H}\Delta\mathcal{H}}(A, T) + \lambda$$

where $\mathcal{H}$ is the hypothesis space, $h$ is a hypothesis function that returns labels $\{0, 1\}$, and $e(T, T)$ and $e(A, T)$ are the generalization errors from passing target domain data $T$ and adversarial data $A$ through a classifier trained on $T$. $d_{\mathcal{H}\Delta\mathcal{H}}(A, T)$ is the $\mathcal{H}\Delta\mathcal{H}$-distance between $T$ and $A$, and measures the divergence between the feature distributions of $A$ and $T$. $e_A(h, h^{'})$ and $e_T(h, h^{'})$ represents the probability that $h$ disagrees with $h^{'}$ on the label of an input in the domain space $A$ and $T$ respectively.

$$d_{\mathcal{H}\Delta\mathcal{H}}(A, T) = \sup_{h, h' \in \mathcal{H}} |e_A(h, h^{'}) - e_T(h, h^{'})|$$

$$d_{\mathcal{H}\Delta\mathcal{H}}(A, T) = \sup_{h, h' \in \mathcal{H}} \left|\mathbb{E}_{X \sim S}[|(h(x) - h^{'}(x)|]\right|$$
$$- \left|\mathbb{E}_{X \sim T}[|(h(x) - h^{'}(x)|]\right|$$

Divergence $d_{\mathcal{H}\Delta\mathcal{H}}$ measures the divergence between feature distributions $A$ and $T$. Higher $d_{\mathcal{H}\Delta\mathcal{H}}$ indicates less shared features between 2 domains. The greater the intersection between feature distributions, the greater the proportion of domain-variant features; one approach to domain adaptation is learning domain-invariant features representations (Zhao et al., 2019) to minimize $d_{\mathcal{H}\Delta\mathcal{H}}$.

| Target Domain | magazine | | | baby | | |
|---|---|---|---|---|---|---|
| Attack Domain | baby | dvd | book | dvd | book | magazine |
| After-attack Accuracy | 0.381 | 0.366 | 0.343 | 0.365 | 0.386 | 0.401 |
| After-defense Accuracy | | | | | | |
| Adversarial training | 0.639 | 0.559 | 0.657 | 0.558 | 0.577 | 0.661 |
| Defensive distillation | 0.549 | 0.561 | 0.597 | 0.588 | 0.629 | 0.577 |
| SharedVocab defense | 0.628 | 0.653 | 0.631 | 0.664 | 0.668 | 0.621 |
| Domain-adapted adversarial training | 0.608 | 0.637 | 0.620 | 0.604 | 0.620 | 0.587 |
| **Learn2Weight** | **0.796** | **0.842** | **0.843** | **0.774** | **0.751** | **0.737** |

Table 3: After-defense accuracy performance of different defensive methods.

**Explaining similarity-domain attacks.** As demonstrated by empirical results, $e(A, T)$ increases in a similarity-based attack setting, and this would arise if $d_{\mathcal{H}\Delta\mathcal{H}}$ increases correspondingly. $d_{\mathcal{H}\Delta\mathcal{H}}$ computes inconsistent labels from inconsistent feature distributions, and attributes the success of the attack to domain-variant features.

*FGSM* and variants adjust the input data to maximize the loss based on the backpropagated gradients of a model trained on $S$. As our pipeline used correctly-labelled sentences before adversarially perturbing them, we can infer that perturbations applied to $S$ were not class-dependent (i.e. the success of the attack is not based on the removal of class-specific features), but class-independent features. It is already difficult for a model trained on $S$ to classify when there is insufficient class-dependent features (hence a high $tf(A, T)$); in a cross-domain setting, it must be even more difficult for a model trained on $T$ to classify given a shortage of domain-invariant, class-dependent features.

$$d_{\mathcal{H}\Delta\mathcal{H}} \leq e(A, T) - e(T, T) - \lambda$$

$$d_{\mathcal{H}\Delta\mathcal{H}} \leq tf(A, T) - \lambda$$

**Explaining Learn2Weight.** $L2W$ minimizes divergence by training on $\{d_{\mathcal{H}\Delta\mathcal{H}}(S_j, T) : \Delta W_{A\_S_j}\}$ pairs, $d_{\mathcal{H}\Delta\mathcal{H}}(S_j, T)$ being reconstructed from the difference between $x_i^{S_j}$ and $x_i^T$. $L2W$ is trained on $\{d_{\mathcal{H}\Delta\mathcal{H}}^{S_j}\}_{j=0}^N : \{\Delta W^{S_j}\}_{j=0}^N$ pairs, such that $\Delta \hat{W}^{S_j} = L2W(d_{\mathcal{H}\Delta\mathcal{H}}^{S_j})$. Intuitively the target model possesses a decision boundary (Liu et al., 2019) to classify inputs based on whether they cross the boundary or not; adversarial inputs have a tendency of being near the boundary and fooling it. Weights transfer learning applies perturbations to the decision boundary such that the boundary covers certain adversarial inputs otherwise misclassified, and in this way blocks transferability. The advantage of training on multiple domains $\{S_j\}_{j=0}^M$ is that the after-$L2W$ divergence between $A$ and $T$ is smaller because $L2W$'s weight perturbations render the decision boundary more precise in classifying inputs.

**Explaining tf-optimization.** We have attributed why adversarial sentences $A$ are computed to be domain-dissimilar despite originating from $S$ due to insufficient domain-invariant, class-dependent features resulting in low $e(A, T)$, i.e. low $tf(A, T)$. To replicate this phenomenon in natural domains, we use $tf$-optimization to iteratively perturb $T$ to increase the proportion of class-independent features. This approximates the real-world similarity-based attack scenario where class-dependent features may be limited for inference. By generating the synthetic data, we are feeding $L2W$ attack data with variations in $d_{\mathcal{H}\Delta\mathcal{H}}$ and class-independent feature distributions. This prepares $L2W$ to robustify weights in $f(T)$ when such feature distributions are encountered.

## 6 EXPERIMENTS

### 6.1 BASELINES

We consider two defense strategies that are empirically effective and are widely used for general black-box adversarial attacks: adversarial training (Goodfellow et al., 2014; Madry et al., 2017) and defensive distillation (Papernot et al., 2016b; 2017). In addition we consider two ablation baselines.

**Defensive distillation:** The high-level implementation of *defensive distillation* (Papernot et al., 2016b; 2017) is to first train an initial model against target domain inputs and labels, and retrieve the raw class probability scores. The predicted probability values would be used as the new labels for the same target sentences, and we would train a new model based on this new label-sentence pair.

| Target Domain | Original Accuracy | Attack Domain | Attack Model After-attack Accuracy | | | | | After-Defense Accuracy | | | | |
|---|---|---|---|---|---|---|---|---|---|---|---|---|
| Target Model: LSTM | | | BERT | LSTM | GRU | CNN | LogReg | **BERT** | **LSTM** | **GRU** | **CNN** | **LogReg** |
| book | 0.880 | dvd | 0.342 | 0.413 | 0.477 | 0.335 | 0.440 | **0.786** | **0.847** | **0.804** | **0.816** | **0.782** |
| | | kitchenware | 0.350 | 0.372 | 0.325 | 0.353 | 0.425 | **0.765** | **0.826** | **0.795** | **0.742** | **0.767** |
| | | electronics | 0.400 | 0.389 | 0.416 | 0.315 | 0.460 | **0.792** | **0.812** | **0.784** | **0.770** | **0.725** |
| dvd | 0.920 | book | 0.326 | 0.434 | 0.479 | 0.383 | 0.490 | **0.816** | **0.795** | **0.824** | **0.804** | **0.794** |
| | | kitchenware | 0.355 | 0.370 | 0.379 | 0.359 | 0.490 | **0.728** | **0.796** | **0.755** | **0.735** | **0.695** |
| | | electronics | 0.387 | 0.377 | 0.332 | 0.348 | 0.455 | **0.825** | **0.836** | **0.812** | **0.834** | **0.796** |
| electronics | 0.910 | book | 0.425 | 0.394 | 0.473 | 0.358 | 0.474 | **0.775** | **0.821** | **0.795** | **0.782** | **0.712** |
| | | dvd | 0.342 | 0.395 | 0.452 | 0.368 | 0.493 | **0.784** | **0.845** | **0.855** | **0.842** | **0.792** |
| | | kitchenware | 0.390 | 0.384 | 0.464 | 0.329 | 0.432 | **0.730** | **0.824** | **0.753** | **0.724** | **0.678** |

Table 4: Learn2Weight comparison against different attack model architectures.

**Adversarial training:** It is shown that injecting adversarial examples throughout training increases the robustness of target neural network models. In this baseline, target model is trained with both original training data and adversarial examples generated from original training data. However, since the adversarial examples are still generated from the target domain, it is unlikely that the method can defend similar-domain attack which is the result of domain-variant features.

**SharedVocab defense.** Given that it is the domain-variant features that cause the success of similar-domain attack, a simple baseline is to remove those words that are *not* in the target domain's vocabulary. This tests whether the effect of perturbing target model weights w.r.t domain-variant features will yield incremental after-attack accuracy in the similar-domain attack setting.

**Domain-adapted adversarial training.** This ablation baseline tests for incremental performance to a baseline defense using domain-variant inputs. We adapt adversarial training to be trained on adversarial sentences from attack domain $S$, whereas the traditional adversarial training generates adversarial samples $X^{adv,T}$ from its training data $T$, this adapted version uses adversarial samples $X^{adv,S}$ generated from $S$.

## 6.2 LEARN2WEIGHT PERFORMANCE

**Defense performance**. We present the results of different defense baselines in Table 3. First, we can see that Learn2Weight achieves the highest after-defense accuracy against the adversarial attack. Take the *magazine* as target domain for example, if the adversary chooses to use *book* data as the attack domain, it would reduce the target model accuracy to 0.343. However, the Learn2Weight method can improve the performance to 0.843, which is a significant and substantial improvement against the attack. This improvements also exist across different target/attack domain pairs. Second, we see that all defense methods can improve the accuracy to some extent which indicates the importance and effectiveness of having robust training for machine learning models. Third, it is interesting to note that two simple baselines SharedVocab defense and Domain-adapted adversarial training yield overall better performance compared to adversarial training and defensive distillation.

**Attack model architectures.** So far, all the results are conducted using the same LSTM as the target/attack model due to simplicity purpose. Here, we keep the target model unchanged, but vary the architecture of the attack model for the generation of adversarial examples. A variation of a Recurrent Neural Network is a *Gated Recurrent Unit (GRU)* network, with 512 GRU cells, 60% dropout and $tanh$ activation function. We have also tested other attack model variants that are commonly-used in sentiment classification, including *Bidirectional Encoder Representations from Transformers (BERT)* (Devlin et al., 2019), *Convolutional Neural Network (CNN)* (Kim, 2014), and *Logistic Regression* (Maas et al., 2011). For both RNN and CNN, we use pre-computed Glove embeddings[2] to encode words. All models are trained with enough epochs after ensuring the model achieved near state-of-the-art validation accuracy before proceeding to tests of adversarial attacks and defenses.

We present the results of different attack model architectures in Table 4. First, similar-domain attack is model-agnostic and it does not require the target and attack model to have identical architectures.

---

[2]Embeddings available from https://nlp.stanford.edu/projects/glove/

We can see that all four attack model architectures are able to reduce the target model accuracy. Second, the results suggest that Learn2Weight is also model-agnostic as it can substantially improve the after-defense accuracy regardless which attack model is used.

## 7 CONCLUSION

In this newly-proposed, empirically-effective similar-domain attack, an adversary can choose a similar domain to the target task, build a substitute model and produce adversarial examples to fool the target model. We also propose a defense strategy, Learn2Weight, that learns to adapt the target model's weight using crafted adversarial examples. Compared with other adversarial defense strategies, Learn2Weight can improve the target model robustness against the similar-domain attack. Our method demonstrates properties of a good adversarial defense, such as adopting defense architectures that adapt to situations/inputs rather than compromising standard error versus robustness error, to leverage class-independent properties in domain-variant text, and factoring in domain similarity in adversarial robustness exercises.

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
