# OpenReview forum: "Learn2Weight: Weights Transfer Defense against Similar-domain Adversarial Attacks"
_ICLR.cc/2021/Conference — Reject_

### Official Review · AnonReviewer1 · 2020-10-26
**Official Blind Review #1**

**Rating:** 3
**Confidence:** 3

**Review:**

In this paper, the authors propose a learn2weight framework to defend against similar-domain adversarial attacks. Experimental studies on Amazon dataset are done to verify the proposed learn2 weight.

The paper is not easy to follow. The presentation and organization should be further improved. Here are the detailed comments:

(1)	Adversarial attacks are widely used in various application domains, e.g., computer vision [ref1] and reinforcement learning [ref2]. It is necessary to discuss with these related works, and highlight the difference and importance of adversarial attack methods on NLP tasks.

[ref1] Adversarial Examples that Fool both Computer Vision and Time-Limited Humans

[ref2] Minimalistic Attacks: How Little it Takes to Fool Deep Reinforcement Learning Policies

(2)	The authors highlight “domain adaptation theory” several times. Please give a clear description on what it is.

(3)	Where is Table 1 used in the main content?

(4)	Regarding definition 2, the following two points are unclear: (1) is f_S(A) the true label of A. Based on the figure 1 (a), only correctly classified source samples are used while the definition does not show this. (2) why f(A,W_T) = f_S(A)? f is the target classifier, are you generating the domain-invariant samples?

(5)	The rationale of similar domain adversarial attack is confused. It is more reasonable to use source data to help generate target adversarial samples X’ which confuse the classifier to deviate the label f(X) \neq f(X’) where X is the original target sample. However, the paper generates source adversarial samples, which naturally may confuse the target classifier due to the domain divergence. It is unclear why and how these source adversarial samples can contribute to the robustness of the target classifier.

(6)	Regarding the accuracy drops in Table 2, it is highly possible caused by the data shift between different domains. How to differentiate the importance of the data shift and adversarial in the accuracy drops?

(7)	The technical part is not easy to follow. The sections 5.1 to 5.3 are not linked well. It is necessary to give more contents on the motivation and flow of these algorithms instead of just putting them in algorithm charts.

(8)	Why target data is used in Algorithm 2 and also transfer loss optimization? In the introduction, target domain information is assumed to be unavailable. Moreover, algorithm 2 is to reduce the domain divergence (if I understand correctly). I am quite curious how the proposed method differentiates from other transfer learning methods.

Update: Thanks for the authors' response. After reading the response and the other reviewers' comments, I think the paper needs to be further improved, and thus I will keep my score.

---

> ### Author Response · Authors · 2020-11-17
> **Clarification on descriptions, domain shift, robustness**
>
> Thank you very much for reviewing our work, we appreciate your time and effort.
>
> **Domain shift**
>
> Table 2 is updated with “unperturbed accuracy”, a metric that measures the accuracy of target model against the original test set of the attack domain (unperturbed). This shows the drop in accuracy attributable to data/domain shift. If we compare this line to the after-attack accuracy (where we apply perturbations), there is a further drop in accuracy, not only from data/domain shift but from further exploitation of the feature distribution of the domain.
>
> **vs other methods**
>
> Other than the benefit of defending against the similar-domain adversarial attack, Learn2Weight also retains both generalization and robustness accuracies. For common domain generalization methods, while they may increase robustness accuracy against foreign domains, they tend to have worsened standard accuracy [2]. In fact we did include a domain generalization defense in our ablation studies (domain-adapted adversarial training), which incorporated domain-adapted adversarial samples during the training of the target model to generalize across diverse domains of inputs; the result remained poor as indicated in Table 3. Extending even further, many methods that incorporate the concept of generalization or model robustification during the training of a model tend to suffer from a weaker standard accuracy at the cost of higher robustness accuracy, including the adversarial training defense [3].
>
> **Natural confusion**
>
> Referring to Table 2 with 2 new rows added, we see that the after-attack accuracy (accuracy of target model against similar-domain adversarial samples) is lower than both intra-attack accuracy (target model against adversarial samples generated from the same target domain) and unperturbed accuracy (target model against unperturbed / non-adversarial test data of attack domain). Unperturbed accuracy accounts for the tendency that domain divergence may naturally confuse the target classifier, but our results with after-attack accuracy shows that applying perturbations on data of divergent domains further weakens model classification accuracy.
>
> **Robustness**
>
> Learn2Weight and incorporating data from foreign domains robustifies the target classifier against adversarial examples crafted from foreign domains. Such adversarial examples can adversarially cross the decision boundary of a model based on its adversarial perturbations as well as domain shift, thus a model with dynamic gradients with prior training on data with perturbations and synthetic domain shift can maintain robustness as well as retain generalizability.
>
> **Table 1**
>
> Table 1 illustrates what adversarial samples with perturbations look like to a reader. Perturbed words are highlighted in blue to indicate that the substitution of those words cause a classifier trained on target domain to misclassify the adversarial sentence, while a classifier trained on the attack domain continues to classify the adversarial sentence correctly. The original sentence from the attack domain is provided, which is not only correctly classified by a model trained on the attack domain but also correctly classified by a model trained on the target domain.
>
> **Definition 2**
>
> For (1), $f_S(A)$ is the adversarial label of A; in the adversarial attack algorithm $Adv(f_S, S)$, perturbations are introduced with respect to the gradients of $f_S$, thus the label of A is not equal to S.
>
> For (2), we have added the condition “during an attack” to indicate that $f(A,W_T) = f_S(A)$ is an intended goal of the attack rather than an explicit objective function for adversarial example generation.
>
> **Algorithm 2**
>
> Algorithm 2 is from the defender’s perspective, not attacker’s perspective. Target domain information is not available to the attacker, but available to the defender because the defender is building models on their own data (target domain data).
>
> References:
>
> [1] Shai Ben-David, John Blitzer, Koby Crammer, Alex Kulesza, Fernando Pereira, and Jennifer Vaughan. A  theory  of  learning  from  different  domains. Machine  Learning,  2010.
>
> [2] Wang, H., Ge, S., Lipton, Z., & Xing, E. (2019). Learning Robust Global Representations by Penalizing Local Predictive Power. NIPS.
>
> [3] Aditi Raghunathan, Sang Michael Xie, Fanny Yang, John C Duchi, and Percy Liang.  Adversarial Training can hurt generalization.arXiv, 2019.

---

### Official Review · AnonReviewer2 · 2020-10-27
**Learn2Weight and similar domains attacks**

**Rating:** 5
**Confidence:** 5

**Review:**

Summary:
This paper is about generating adversarial examples for some target model and protecting from such attacks.  Authors consider a setting when an adversary has access to some "similar to target " domain data, and can use this data to generate a surrogate model. Using this surrogate model an adversary can generate adversarial examples, that apparently also fool the target model. Then authors also propose a defense mechanism from this type of attack, Learn2Weight. This is a learnt network that, for a given example, returns perturbation of weights to the target model which will be applied to the target before inference. This model is trained by a defender on synthetic domains generated as perturbations to the target data

Overall, this type of an attack is interesting. The paper is well organized and written, and easy to follow. Enough background is given for a reader to follow without the need to research around or going to appendix. Well done on clarity!
 I do have a problem understanding how effective this attack is (compared to other blackbox attacks) and how the proposed defense compares to standard domain generalization methods like learning domain invariant features.

1) One concern I have is about practicality an availability of such "similar" domains. For testing authors used Amazon multi-domain sentitment classification, where domains are easily available. But how would you attack a pre-trained Imagenet for example?
- What domains are similar?
- and further more, how much data for these similar domains you need to have to train a good enough surrogate model?
- Also you don't really have a way to calculate that your data is close to the actual target data.
2) Definition 2: f(A, W_T) = f_S(A) requires an access to your model f, so I would not call this type of attack "without access to the target model"
3) How does this attack compares to any other black box attack that uses target model? It really should be in Table 2. If other attacks are able to make target model performance worse than this type of attack, it is of less value to defend from a weaker attack
4) Algo 3 - what are the adversarial perturbations you are talking about?
5) I am not sure algorithm 2 is the best way of doing it? Why not to try any of domain generalization techniques (e.g. train on all domains with an adversarial head tries to distinguish between domains, or MMD or whatever). May be this way you won't need Learn2Weight model at all (since you are already learning domain invariant features only)

Minor:
- Table 2: What are u boldening ? I would expect the bolden result to be per source model (book) and the worse performance you get (so dvd attack gives the lowest after attack accuracy). You are boldening "baby", which is the weakest domain (on which your attack mode is trained) for an attack.
- Algo 2 Compute weights of f trained on TY=W_T-W_T (just assign 0s?)

---

> ### Author Response · Authors · 2020-11-17
> **Response on practicality, methods**
>
> Thank you very much for reviewing our work, we appreciate your time and effort.
>
> **Domain generalization**
>
> Other than defending against similar-domain adversarial attack, Learn2Weight retains both generalization and robustness accuracies. For common domain generalization methods, while they increase robustness accuracy against foreign domains, they have worsened standard accuracy [6]. We do include a domain generalization defense (domain-adapted adversarial training, Table 3). Many methods that incorporate generalization or model robustification during the training of a model tend to suffer from a weaker standard accuracy at the cost of higher robustness accuracy, including the adversarial training defense [7].
>
> **ImageNet**
>
> The methods to obtain “similar” domains may vary, but the pipelines presented still hold (e.g. the defender is assumed to have no other domains other than their own target domain, and is able to generate synthetic domains via transfer loss optimization). For ImageNet, some methods could be (1) Obtain labelled datasets from other domains, such as paintings [3] or comics [4], or (2) Obtain dataset with similar class labels as ImageNet and apply style transfer [5], or (3) `Utilize transfer loss optimization to synthesize similar domain. The empirical definition of whether any two domains are considered “similar” is based on transfer loss, which is task-agnostic.
>
> **Domain similarity**
>
> Domain similarity is measured by 2 metrics (section 3.3). For SharedVocab, a higher degree of overlapping vocabulary implies the two domains are more similar. For transfer loss, lower loss indicates higher similarity. We can determine which attack domain data are similar to the target domain data. The similar attack-target domain pairs include {magazine-book, dvd-book, dvd-magazine, book-magazine}, while dissimilar pairs include {baby-book, baby-magazine, dvd-baby, book-baby, magazine-baby}.
>
> **Weight assignment**
>
> We retain consistency with $dW = W_{Sj} − W_T$ in Algo 2. For a reader, it is intuitive to infer 0s from $W_T-W_T$, but not so to infer $W_T-W_T$ from 0s.
>
>  **Training surrogate**
>
> Each product category contains 2,000 original sentences (1,000 positive-labelled, 1,000 negative-labelled). We perform a 80-20 train-test split, thus using 1,600 sampled sentences to construct the surrogate model.
>
> **Perturbations**
>
> The adversarial perturbations being applied to target domain sentences in Algo 3 are word-level substitutions by FGSM [1, 2].
>
> An adversarial example can be mathematically denoted as $\{ x_{i}^{adv} \}_{i=0}^{n}$. Each sentence $X$ consists of $n$ words $\{x_1, x_2,..., x_n\}$. The degree of perturbation parameter $\epsilon$ changes the value assigned to each word $x_i$ into $x_i^{'}$ and the proportion of $X$ to be perturbed, for each word at position index $i$. What constitutes a perturbation using the FGSM algorithm could be represented as:
>
> $$ x_{i}^{adv} = x_{i} + \epsilon sign (\nabla_x * J(\theta, x_i, y_i)) $$
>
> **Definition 2**
>
> We have added the condition “during an attack” to avoid confusion. $f(A, W_T) = f_S(A)$ is an intended goal of a similar-domain adversarial attack, but is not an objective function to generate adversarial examples (as we do not have access to the target model); it is conveying that $f_S(A)$ would tend towards $f(A, W_T)$ for a sample A to be successful.
>
> **bold**
>
> They refer to dissimilar pairs to show the correlation between transfer loss and after-attack accuracy.
>
> **vs other attacks**
>
> We have updated Table 2 with values for the intra-attack accuracy. This is the standard scenario when an attacker has access to target domain data and generates adversarial examples from target domain data. We highlight that the marginal after-attack accuracy difference between using identical datasets vs similar domains is caused by the adversarial properties of domain divergence and is one of the primary interests of the paper.
>
> `References:
>
> [1] Ian J Goodfellow, Jonathon Shlens, and Christian Szegedy. Explaining and harnessing adversarial examples. arXiv, 2014.
>
> [2] Puyudi Yang, Jianbo Chen, Cho-Jui Hsieh, Jane-Ling Wang, and Michael I. Jordan. Greedy attack and gumbel attack: Generating adversarial examples for discrete data, 2018
>
> [3] Painter by Number. https://www.kaggle.com/c/painter-by-numbers/data. Kaggle, 2017.
>
> [4] Cenk Bircanoglu. https://www.kaggle.com/cenkbircanoglu/comic-books-classification. Kaggle, 2017.
>
> [5] Robert Geirhos, Patricia Rubisch, Claudio Michaelis, Matthias Bethge, Felix A. Wichmann, & Wieland Brendel (2019). ImageNet-trained CNNs are biased towards texture; increasing shape bias improves accuracy and robustness. ICLR.
>
> [6] Wang, H., Ge, S., Lipton, Z., & Xing, E. (2019). Learning Robust Global Representations by Penalizing Local Predictive Power. NIPS.
>
> [7] Aditi Raghunathan, Sang Michael Xie, Fanny Yang, John C Duchi, and Percy Liang.  Adversarial Training can hurt generalization.arXiv, 2019.

---

### Official Review · AnonReviewer5 · 2020-11-07
**An idea is presented, but the meaningful evaluation is missing**

**Rating:** 4
**Confidence:** 3

**Review:**

Summary:

The paper considers the adversarial attacks via a surrogate model constructed using data from a different domain. The authors propose a defense from such attacks by a special kind of adversarial training inspired by the idea of domain adaptation. The idea can be useful but raises a lot of questions, especially when looking at the evaluation of the proposed approach.

##########################################################################

Reasons for score: I vote for a reject, as some findings are intriguing, while the experimental results are questionable.

The first major concern is, why do authors consider NLP models and attacks in the paper? It is much easier to work with Image datasets, and if the general idea is new, I suggest to start from this point to verify that the considered domain adaption works well in this scenario.

Also, the proposed attack is not new. It is just a surrogate model attack but using a surrogate model training on the data from a different domain (as the authors suggest due to the unavailability of the initial domain data). Also, for this new attack, the authors don't compare a surrogate model attack trained using the same domain data, which would be interesting to compare.

The authors use only one dataset, which is a bit strange for modern papers. For this dataset, they don't provide a full study, limiting the scope of experiments to particular pairs of source-target domains. From the paper, it is not clear how widely applicable are obtained results.

The comparison is not full. There are a lot of options to be tuned for alternative approaches like Adversarial training or other defenses. The hyperparameter selection for them has a crucial effect on their success. So, we can't say that the proposed approach works better than others.

#########################################################################

Major concerns:

* Only one dataset considered. I think that the inclusion of additional datasets (at least three) would improve the paper and make the conclusion by the authors more solid
* Usage of surrogate models trained on other dataset is not new for general adversarial attacks [1 (mentioned in the paper), 2] and for adversarial attacks in NLP [3]
* LSTM is not the state of the art model for the processing of NLP data
* 4.2. what attack do you use? not explicitly specified. so the results can't be verified by replication of the described experiments
* Table 2 will benefit from adding after-attack accuracy for the original domain. If it is similar to the presented accuracies, then why bother with a new method?
* Table 3 comparison is not fair, as we have no details about training for each approach, e.g. we don't know how many additional examples we add during adversarial training. Also note, that the state-of-the-art for adversarial training is different from described in the paper. See [4, 5]
* Table 4 After-Defense Accuracy for what model is presented? because it should be different for LSTM/GRU/CNN attack model
* Tables 2,3,4 - I suggest to keep the list of pairs (target domain, substitute domain) similar for all tables to be sure that the presented examples are not cherry-picked (also, please consider running your approach on all pairs (target domain, substitute domain) and aggregating all these results)
* Domain adaptation models, from my experience, are not easy to train. It is interesting to access the quality of the models for different runs of Learn2Weight (is it stable? etc.)


1. Nicolas Papernot, Patrick McDaniel, and Ian Goodfellow. Transferability in machine learning: from
phenomena to black-box attacks using adversarial samples. arXiv preprint arXiv:1605.07277,
2016a.
2. Cheng, S., Dong, Y., Pang, T., Su, H., & Zhu, J. (2019). Improving black-box adversarial attacks with a transfer-based prior. In Advances in Neural Information Processing Systems (pp. 10934-10944).
3. Fursov, I., Zaytsev, A., Kluchnikov, N., Kravchenko, A., & Burnaev, E. (2020). Differentiable Language Model Adversarial Attacks on Categorical Sequence Classifiers. arXiv preprint arXiv:2006.11078.
4. Shafahi, A., Najibi, M., Ghiasi, M. A., Xu, Z., Dickerson, J., Studer, C., ... & Goldstein, T. (2019). Adversarial training for free!. In Advances in Neural Information Processing Systems (pp. 3358-3369).
5. Aleksander Madry, Aleksandar Makelov, Ludwig Schmidt, Dimitris Tsipras, and Adrian Vladu.
Towards deep learning models resistant to adversarial attacks. ICLR, 2017.

#########################################################################

Proposed minor improvements:

Table 1: demonstrates one example that breaks the semantics of the attacked sentence. Can you provide good examples of why your approach work?
Definition 1: is not a definition, is X one instance or many instances? in this definition also not specified that X and X' should be similar
Equation 1: why you avoid standard number of equations \begin{equation} \label{eq:sample_equation} sample text \end{equation}?

---

> ### Author Response · Authors · 2020-11-17
> **Clarifications on contributions, experiments, rationale**
>
> Thank you very much for reviewing our work, we appreciate your time and effort.
>
> **Why NLP**
>
> (1) Interpretability: Domain differences between 2 datasets of images could be attributed to many different known or unknown factors [6] such as scene, intra-category variation, object location and pose, view angle, resolution, motion blur, scene illumination, background clutter, camera characteristics, etc. In Amazon multi-domain dataset, there are fewer axes of domain shift. The perturbations and feature distributions can be visualized and interpreted. Metrics such as transfer loss and sharedvocab can confidently ascertain whether 2 domains are similar or not, which is extremely important to support findings of this paper.
>
> (2) Feasibility: Fewer image datasets for domain adaptation [7, 8, 9, 10, 11] constrain the number of attack-target domain pairs (literature increasing domain synthesis [12, 13], while a practical attack needs natural domains).
>
> **1 dataset, particular pairs**
>
> The rationale is the Amazon reviews dataset, with 25 sub-datasets of product categories, is sufficiently diverse. There is literature [4, 5] whereby findings also solely rely on this large, diverse and comprehensive dataset. They also render a few pairs so that the results can be easily interpreted by readers, and we have been consistent with the style of the field in using Amazon reviews dataset to study domain adaptation.
>
> **Novelty**
>
> Our contributions can be supported by Section 5.4. We argued the success of a similar-domain adversarial attack is not hinged on the use of a surrogate model (which we use as an example implementation of the attack), but instead reliant on a shortage of domain-invariant, class-dependent features.
>
> **SOTA**
>
> We intend to demonstrate the effectiveness against a spectrum of attack-defense configurations. In practice not all defenders use state-of-the-art models or defensive parameters, and we use the source methods that most state-of-the-art methods are based on. For completeness, we add Transformer models, and have performed experiments on this and updated Table 4.
>
> **Attack used**
>
> Unless otherwise specified, we used the most basic gradient based method FGSM [3]; refer to Section 3.1.
>
> **Defense parameters**
>
> Adversarial training: Based on Goodfellow et al., 2014 [2], we generate adversarial samples from the target domain training set (800 positive, 800 negative) using FGSM [3], for each epsilon from 0.0 to 1.0 with step 0.1. For each training iteration, we randomly sample 100 samples from each epsilon (i.e. 800 adversarial samples), i.e. training set size per iteration is 2,400. For domain-adapted adversarial training, we use sentences from synthetic attack domain S.
>
> Defensive distillation: The target domain used has 1600 samples, used for generating the class probability scores and training the model (64 LSTM cells, 80% dropout, sigmoid activation).
>
> SharedVocab defense: This defense generates a dictionary from the target domain text corpus, and for each incoming sentence from the attacker, words not found in the target domain dictionary will be replaced with UNK tokens [1].
>
> **Value changes**
>
> Updated: Table 1, Table 4, Table 2, Definition 1
>
> References:
>
> [1] Luong T, Sutskever I, Le Q, Vinyals O, Zaremba Wojciech. (2015). Addressing the Rare Word Problem in Neural Machine Translation. ACL.
>
> [2] Goodfellow IJ, Shlens J, Szegedy C. Explaining and harnessing adversarial examples. arXiv, 2014.
> [3] Alexey Kurakin, Ian Goodfellow, and Samy Bengio. Adversarial machine learning at scale. arXiv, 2016.
>
> [4] Glorot, X., Bordes, A., & Bengio, Y. (2011). Domain Adaptation for Large-Scale Sentiment Classification: A Deep Learning Approach. ICML.
>
> [5] Wanyun Cui, Guangyu Zheng, Zhiqiang Shen, Sihang Jiang, & Wei Wang (2019). Transfer Learning for Sequences via Learning to Collocate. ICLR.
>
> [6] A. Torralba and A. A. Efros, "Unbiased look at dataset bias," CVPR 2011.
>
> [7] J. Deng, W. Dong, R. Socher, L.-J. Li, K. Li, and L. FeiFei. ImageNet: A Large-Scale Hierarchical Image Database. 2009.
>
> [8] M. Everingham, L. Van Gool, C. K. I. Williams, J. Winn, and A. Zisserman. The pascal visual object classes (voc) challenge. IJCV 2010.
>
> [9] L. Fei-Fei, R. Fergus, and P. Perona. Learning generative visual models from few training examples: An incremental bayesian approach tested on 101 object categories. CVPR, 2004.
>
> [10] B. C. Russell, A. Torralba, K. P. Murphy, and W. T. Freeman. LabelMe: a database and web-based tool for image annotation. 2008.
>
> [11] J. Xiao, J. Hays, K. Ehinger, A. Oliva, and A. Torralba. Sun database: Large-scale scene recognition from abbey to zoo. CVPR, 2010.
>
> [12] A. Atapour-Abarghouei and T. P. Breckon. Real-Time Monocular Depth Estimation Using Synthetic Data With Domain Adaptation via Image Style Transfer. CVPR, 2018.
>
> [13] K. Bousmalis, N. Silberman, D. Dohan, D. Erhan, and D. Krishnan. Unsupervised Pixel-Level Domain Adaptation With Generative Adversarial Networks. CVPR, 2017.

---

### Decision · Program_Chairs · 2021-01-07
**Final Decision**

**Decision:**

Reject

**Comment:**

The submission considers a new attack model for adversarial perturbation in a framework where the attacker has neither access to the trained model nor the data used for training the model. The submission suggests a"domain adaptation inspired attack": learn a different model on a similar domain and generate the adversarial perturbations using that model. The authors then also develop a defense for this type of attack and provide some empirical evaluations of the resulting losses on a few NLP benchmark datasets.

The paper refers to the literature on domain adaptation theory to motivate their suggested defense, but this analysis remains on an intuitive (rather than a formally rigorous) level. Furthermore, the empirical evaluation does not compare to a variety of attacks and the defense is only evaluated with respected to the self-suggested attack. This is a very minimal bar for a defense to meet.

The reviewers have criticized the submission for the rather minimal extend of empirical evaluation. Given that the submission also doesn't provide a sound theoretical analysis for the  proposed attack and defense, I agree with the reviewers that the submission does not provide sufficient novel insight for publication at ICLR.

In contrast to some of the reviewers, I do find it legitimate (and maybe recommendable even) to focus on one chosen application area such as NLP. I don't see a requirement to also present experiments on image data or re-inforcement learning applciations. However, I would recommend that the authors highlight more explicitly what general lessons a reader would learn from their study. This could be done through a more extensive and systematic set of experiments or a through analysis in a well defined theoretical framework.